# Rotavirus Sickness Symptoms: Manifestations of Defensive Responses from the Brain

**DOI:** 10.3390/v16071086

**Published:** 2024-07-06

**Authors:** Arash Hellysaz, Marie Hagbom

**Affiliations:** Division of Molecular Medicine and Virology, Department of Biomedical and Clinical Sciences, Linköping University, 581 85 Linköping, Sweden; arash.hellysaz@liu.se

**Keywords:** rotavirus, gastroenteritis, CNS, sickness symptoms, behavioral responses, evolution

## Abstract

Rotavirus is infamous for being extremely contagious and for causing diarrhea and vomiting in infants. However, the symptomology is far more complex than what could be expected from a pathogen restricted to the boundaries of the small intestines. Other rotavirus sickness symptoms like fever, fatigue, sleepiness, stress, and loss of appetite have been clinically established for decades but remain poorly studied. A growing body of evidence in recent years has strengthened the idea that the evolutionarily preserved defensive responses that cause rotavirus sickness symptoms are more than just passive consequences of illness and rather likely to be coordinated events from the central nervous system (CNS), with the aim of maximizing the survival of the individual as well as the collective group. In this review, we discuss both established and plausible mechanisms of different rotavirus sickness symptoms as a series of CNS responses coordinated from the brain. We also consider the protective and the harmful nature of these events and highlight the need for further and deeper studies on rotavirus etiology.

## 1. Introduction

Rotavirus infection is one of the leading causes of pediatric viral gastroenteritis. It is highly contagious and has been estimated to infect every newborn child in the world at least once before the age of five [1]. In healthy older children and adults, rotavirus infections can be mild or even asymptomatic. Therefore, its role as a pathogen in adults has been vastly underappreciated [2]. Nonetheless, it was recently found that in 1/3 of the families of children hospitalized with rotavirus, a caregiver also becomes ill and suffers from gastroenteritis [3]. Outbreaks among adults have also been reported from various parts of the world recently [4,5,6].

The hallmark symptoms of rotavirus infection are diarrhea and vomiting, which in children can lead to dehydration associated death if left untreated [7]. As such, most studies on rotavirus have focused on the intestinal mechanisms of diarrhea. Conversely, other relevant and common sickness symptoms like fever, nausea, malaise, headache, abdominal discomfort, myalgias, fatigue and loss of appetite [8] have received far less attention. Severe adult rotavirus gastroenteritis with multi-organ failure and critical management have in rare cases also been reported [9], further highlighting the complexity of rotavirus etiology.

Most of our knowledge of rotavirus pathophysiology comes from animal studies, which have established that the infection is commonly restricted to mature enterocytes of the small intestine [10,11]. Extraintestinal spread only occurs in rare cases [12,13,14]. Contrary to what could be expected from the symptomology, infection does not, however, induce a distinctive cytokine response [10,15,16,17,18]. Instead, a growing body of evidence points towards gut–brain crosstalk driving the defensive response against rotavirus infection [19].

The involvement of the brain in mediating rotavirus sickness symptoms has only recently started to be investigated. This includes vomiting through activation of the vomiting center in the medulla oblongata of the brainstem [20], but also increased intestinal motility by downregulation of the sympathetic nervous system, specifically to the ileum, but not the duodenum or jejunum [21]. These new findings together with previous knowledge about how sickness symptoms generally occur suggest the central nervous system (CNS) as a driving force of defensive responses and consequently sickness symptoms during rotavirus infection.

In this review, we focus on the clinically established rotavirus symptomology beyond diarrhea. Although the precise mechanisms of most rotavirus sickness symptoms have not been elucidated, we discuss them as results of a series of coordinated systemic events, that are likely synchronized by the CNS, against the pathogen, to increase the survival of the individual host as well as the collective group. We also consider evolutionary perspectives and the advantages and the trade-offs of these defensive strategies, and discuss the harmful and protective nature of rotavirus sickness symptoms.

## 2. Gut–Brain Crosstalk

The idea that the gut and the brain are connected has been a medically accepted subject for a long time. Already in the 10th century, the great Iranian physician Ibn Sina, also known as Avicenna (980-1037 AD), anatomically described the nervous connection (Figure 1) of the peritoneum to the spine and the brain [22]. In his extended medical encyclopedia Qānūn fī al-Tibb (the Canon of Medicine), which was written in 1025 AD and used as the main medical reference textbook in Europe until the 17th century [22,23,24,25], Ibn Sina explained how imbalance in the gut could, via specific direct and indirect pathways, be relayed to the brain and contribute to the pathogenesis of a number of diseases including headache, melancholia, nausea, bowel incontinence [i.e., diarrhea], and vomiting [26].

Many aspects of these early medieval descriptions persist in current anatomy [27]. Molecular biological and mechanistic knowledge about the interconnectivity of the gastrointestinal tract and the CNS through direct and indirect ascending and descending pathways (Figure 2) have been vastly extended in the last century, and the concept is well established in modern medicine [28,29]. These pathways can explain how viral gastroenteritis can cause various sickness symptoms. Surprisingly, gut–brain crosstalk has remained poorly studied in the context of rotavirus pathogenesis [19]. 

### 2.1. Ascending Pathways

Information to the CNS about the presence of a gastrointestinal pathogen can reach the brain through three major types of pathways, which consist of (a) electrical nervous signaling, (b) direct invasion by the pathogen into the brain, and (c) chemical signaling by means of various messenger molecules like cytokines, hormones, peptides, and toxins, which are transported to the brain in the vascular or lymphatic pathways [19]. Importantly, these pathways operate at different speeds and temporal resolutions. They can occur at different time points and last for shorter or longer periods of time over the course of the disease. While nervous signaling provides instant communication that could elicit a quick and short defensive response and consequent sickness symptoms, like vomiting, within a short timeframe after infection, circulating messenger molecules like toxins, cytokines, and hormones, which could affect multiple organs and for instance alter feeding behavior [30], operate at a slower rate but persist for longer [19].

Of course, multiple ascending and local pathways could also interact to create the complex rotavirus pathophysiology that is clinically observed. For instance, the rotavirus non-structural protein 4 (NSP4) has been found to stimulate enterochromaffin (EC) cells of the small intestine and induce local release of the neurotransmitter serotonin (5HT), which activates the vagus nerve to elicit vomiting [20,31]. Released serotonin is also involved in local regulation of intestinal motility by activating primary afferent nerves of the myenteric plexus, which stimulate the nerves of the submucosa plexus to release vasoactive intestinal peptide (VIP) from nerve endings adjacent to crypt cells [32,33]. 

#### 2.1.1. Nervous Signaling from the Gastrointestinal Tract

The vagus nerve, extending from its origin in the brainstem, extensively innervates different parts of the gastro-intestinal tract and provides nervous feedback from the digestive system to the brain. This includes information from the enteric nervous system (ENS) as well as direct sensory information from mechano-, chemo-, and tension receptors [34]. The cell bodies of sensory afferents to the small intestine are located in the nodose ganglion and project directly into the nucleus of the solitary tract (NTS).

There are similar spinal ascending pathways that directly project into other brain areas, including the parabrachial area, the hypothalamus, and the amygdala [35]. The presence of an infectious pathogen in the intestines can therefore be detected through multiple nervous pathways and rapidly conveyed to the CNS for further processing. This also includes possible interactions between the pathogen and the gut microbiome, which could elicit nervous signaling to the CNS [36,37]. Locally produced cytokines can also activate primary afferent nerves, such as the vagal nerves, during abdominal and visceral infections [38].

#### 2.1.2. Direct Invasion of the Brain

Both viremia and extraintestinal infections of rotavirus in the liver, lungs, and kidneys have been clinically reported [12]. In a prospective study of acute encephalitis in children from Sweden, rotavirus RNA could be identified via PCR in stool samples in 10% of the patients in the cohort [39]. Encephalopathy [40], acute cerebellitis [41], and other CNS infection-associated complications have also been found to be concurrent with rotavirus gastroenteritis. Based on these reports, direct invasion by rotavirus of the brain has been theorized [42]. Nonetheless, the underlying disease causalities in these reports remain unknown, and to the best of our knowledge, no substantiated evidence of rotavirus infection of the human brain has been provided yet.

Recent immunohistochemical data from mice clearly show that the epidemic diarrhea of infant mice (EDIM), rotavirus is restricted to the gastrointestinal tract up to 72 h post-infection [43] and does not extend to the brain even though altered brain activity is observed earlier [21]. Early studies performed by L. M. Kraft on EDIM before it was recognized as a rotavirus [44] did however identify infectious virus in the lungs, liver, spleen, kidneys, bladder, brain, and blood by 72 h post-infection [45,46,47]. Since these organs are highly vascular, it was assumed that the presence of virus reflected the presence of blood rather than actual infection of these organs. Nonetheless, extraintestinal rotavirus infection in humans remains a controversy and is also likely to be attributed to premorbid characteristics like immunodeficiency [45] or hereditary factors [39].

#### 2.1.3. Chemical Signaling

Cytokines, hormones, and peptides function as signaling molecules and can be transported in the vascular and lymphatic systems from peripheral organs to the CNS. The CNS constantly monitors for new or altered signaling molecules. Any induced alteration caused by pathogenic infection can potentially elicit a CNS response. Cytokines are a response to infection, and the main cytokines involved in sickness responses are the pro-inflammatory cytokines IL-1 and TNF-α [38]. Moreover, anti-inflammatory cytokines regulate the intensity and duration of sickness behavior. 

Infection can affect the ENS and potentially alter regulating gut and brain peptides and hormones like peptide YY (PYY), glucagon-like peptide-1 (GLP-1), cholecystokinin (CCK), leptin, or ghrelin [30]. Cytokines have, through both direct and indirect actions, a profound effect on systemic metabolism and the development of sickness behaviors [48].

### 2.2. Descending Pathways

The brain can coordinate defensive responses and output to the periphery through either neural or humoral pathways [30]. The sympathetic and parasympathetic branches of the autonomic nervous system are the main neural outputs that provide the brain direct regulatory access to peripheral organs, including the gastrointestinal tract [49]. These two antagonistic systems operate like “gas and brakes” for a multitude of peripheral organs and systems.

Sympathetic activation increases, for instance, intestinal motility, blood sugar levels, heart rate, blood pressure, and breathing rate, which can make one ready to combat or fly from immediate danger. Activation of the parasympathetic nervous system, on the other hand, has opposing effects and is therefore known to drive the rest and digest conditions [50]. Importantly, stable conditions are maintained through proper balance between the two systems. For instance, increased intestinal motility can be achieved by either increasing the sympathetic or reducing the parasympathetic signaling to the intestines. The autonomic nervous system is an important part of an organism’s survival in managing external danger, but also in combating internal pathogens [51]. 

The neuroendocrine system constitutes the main humoral output of the brain. It is mainly regulated by the neurons of the hypothalamus, which, either directly or indirectly through specialized endocrine cells of the pituitary, release hormones into circulating blood where they can reach the entire body [52].

The neuronal and humoral pathways provide the brain with great versatility to coordinate defense mechanisms against infections. While neuronal pathways provide a near-instant signaling that can be directed to specific organs, the humoral pathways provide the means to, albeit at a slower speed, direct broad systemic signaling to the entire body. It is also well accepted that the CNS is, through both neural and humoral pathways, involved in complex bidirectional communication with the immune system and can control peripheral immunity [53].

## 3. Central Coordination of Defense

Brain areas that receive primary ascending afferents to drive the peripheral inputs to other brain regions are known as first-order nuclei [54]. The NTS is therefore often referred to as a first-order nuclei for the vagus nerve [30]. Conversely, the area postrema (AP), which contains a chemoreceptor trigger zone, is the first-order nuclei to detect circulating emetic agents [55]. From these entry points, the signal can be relayed to a multitude of other brain areas, including the amygdala, the thalamus, and the hypothalamus [49], which are involved in regulating various outputs including mood [56], behavior, and homeostatic balance.

Modulation of various brain structures following rotavirus infection in mice has recently been reported. This includes increased cFos expression in the NTS and AP [20] as well as reduced phosphorylated signal transducer and activator of transcription 5 (pSTAT5) expression in the bed nucleus of the stria terminalis (BNST) [21]. The common denominator for these structures is their role as first-order nuclei. 

The NTS in the dorsal medulla is a major input for vagal afferents from various organs including the gastrointestinal tract. It is part of the dorsal vagal complex and responsible for triggering vomiting [49]. Furthermore, the NTS also projects into a large number of other regions, including the hypothalamus, from which several other rotavirus sickness symptoms like fever, fatigue, sleepiness, stress, and loss of appetite, could arise. Similarly, the BNST receives direct vagal afferents from the periphery [35]. It is a center of integration for limbic information and highly associated with reward, stress, and anxiety. Furthermore, the BNST acts as a relay to the hypothalamic–pituitary–adrenal axis, which regulates acute stress response. 

The AP is a circumventricular organ located caudal to the floor of the fourth ventricle in the medulla [55]. It is highly vascular, lacks a blood–brain barrier, and can sense circulating chemicals in the blood and the cerebrospinal fluid. It projects into various other brain regions including the NTS.

How the signal that arises from rotavirus infection propagates from these nuclei has not been elucidated yet, and systemic investigation of the brain during rotavirus infection is lacking. Nonetheless, these three nuclei together can potentially relay the signal from a rotavirus infection to both neural and hormonal pathways. This does, however, not exclude the possibility of the involvement of other first-order nuclei in rotavirus pathogenesis.

## 4. Sickness Symptoms

The feeling of sickness is not easily measurable and sometimes vaguely defined as the perception of not feeling well. Sickness behaviors follow sickness feelings and include lethargy, social withdrawal, depression, and reduced exploration, but also loss of appetite, sleepiness, and hyperalgesia [57]. It is now abundantly clear that sickness is an active process and not merely a passive consequence of systemic infections [58]. Inducing sickness feelings and monitoring behavioral responses in infants and children are not ethically defensible; however, outbreaks and volunteer studies in adults have shown that sickness symptoms and behaviors occur during rotavirus infection [2].

In a study of 18 rotavirus infected adult volunteers, Kapikan et al. [59] found that 22% develop sickness feelings and 28% shed rotavirus. Observed symptoms among the subjects included diarrhea (22%), vomiting (11%), headache (22%), anorexia (22%), malaise (17%), abdominal cramping (11%), and elevated body temperature (17%). In another study of 83 college students from a rotavirus outbreak [60], the vast majority of the subjects suffered from diarrhea (93%), abdominal pain or discomfort (90%), loss of appetite (83%), and nausea (81%). Furthermore, more than 50% showed the following symptoms: fatigue, vomiting, headache, myalgia, chills, and low-grade fever.

### 4.1. Diarrhea

Diarrhea is one of the hallmark symptoms of rotavirus infection [17]. It is an evolutionarily preserved defense mechanism [61] for effectively driving pathogenic clearance from the lower gastrointestinal tract [62], but simultaneously also very costly to the host, as it can rapidly cause malnutrition and fatal dehydration if sustained over a long period of time [63]. Diarrhea is very effective when it comes to flushing substances from the intestinal lumen, but less so when it comes to getting rid of intracellular pathogens like viral agents, which replicate inside the cells. Nonetheless, reduction of free infectious virus as well as viral toxins from the lumen should some extent be beneficial to the host. In infants and young children, who are less resilient to loss of electrolytes and fluid imbalance, diarrhea is, however, more often harmful. 

The onset of rotavirus diarrhea is between 24–48 h post-infection and has been found to last on average for 6 days among Swedish children [64]. From a cross-sectional study including five hospital-based studies, diarrhea average from the time of presentation ranged from 2.3 to 7.4 days [65]. 

Rotavirus diarrhea was initially regarded to be osmotically driven and emerging by excessive fluid and electrolyte loss due to malabsorption [66,67]. This model did not, however, provide a full explanation for rotavirus etiology. In 1999, the NSP4 enterotoxin was shown to participate in the diarrhea response to rotavirus [68], and in 2000 Lundgren et al. showed for the first time that rotavirus infection induce activation of the enteric nervous system [18]. Today it is well accepted that the underlying mechanisms of rotavirus diarrhea are multifactorial and involve activation of the enteric nervous system (ENS) [18,69,70]. Although the ENS can work independently and regulate many of the intestinal functions such as motility and secretion, there is an autonomic control by the brain to ensure regulation of intestinal homeostasis [71]. Recently, it has been shown that following infection, descending sympathetic nerves from the CNS participate in rotavirus diarrhea by increasing ileal motility [21]. This increase was observed in mice 16 h post-infection, which is before the onset of diarrhea. The central pathways that induce this autonomic response during rotavirus infection have, however, not yet been studied.

### 4.2. Vomiting

Evolutionarily, our physiological capabilities developed at a time when one had to race across the savannah to hunt or catch the next meal [72]. It is only during the last century in modern times that humans have been surrounded by a plethora of food which is highly nutritious and largely free from pathogens and toxins. During evolution, vomiting was probably the most important defense mechanism against food poisoning and uptake of noxious chemicals. Vomiting rapidly and effectively empties the stomach and eliminates the risk of gastric uptake of harmful particles [72]. The cost of falsely inducing vomiting, when no harmful agent is present, is only a few calories [73]. However, the penalty for a single miss, when harmful toxins or pathogens are present in the food, is extremely high and might even lead to death.

Vomiting is commonly the first symptom of rotavirus infection and occurs simultaneously with fever [74]. In a study of rotavirus-infected children, vomiting was the onset symptom in 55% of the patients, preceding diarrhea by 24 h [64]. Interestingly, vomiting usually lasts for only 2 days [64,75], even though virus replication continues for longer and shedding persists for up to 10 days post-infection [76]. It is thus less likely for circulating emetic agents to drive rotavirus vomiting through activation of chemoreceptors in AP. The symptomology instead suggests direct nervous signaling to drive rotavirus vomiting.

Indeed, investigations in mice have identified activation of the NTS following rotavirus infection, and it is now well accepted that rotavirus vomiting occurs through serotonergic activation of the vagus nerve, which relays the signal to the NTS [20]. The serotonergic surge in the intestines is caused by the release of rotavirus NSP4 enterotoxin, which induces calcium increase in the enterochromaffin cells of the intestines, and therefore the release of serotonin [20]. How the body quickly moderates this NSP4-5-HT vagal pathway to alleviate rotavirus vomiting remains elusive. The anti-emetic 5-HT_3_ receptor antagonist ondansetron, which blocks 5-HT binding to vagal afferent nerves, is clinically used to viral gastroenteritis, although not an established treatment alternative for gastroenteritis. There have been clinical trials of ondansetron use against viral gastroenteritis [77,78], but more studies are needed to confirm positive or negative effects from inhibiting the host vomiting response. 

### 4.3. Fever

Fever is a common response to infection that has been conserved in vertebrates for over 600 million years [79]. Fever even occurs in cold-blooded vertebrates like reptiles and fish, which raise their core temperature during infection by altering their behavior and seeking warmer environments, despite the risk of predation.

Fever is very common during rotavirus infection [64,75,80]. Rotavirus illness usually begins with acute onset of fever and vomiting, and about 30–40% of children may experience temperatures above 39 °C [74]. In a prospective study of acute gastroenteritis in Swedish children, about 84% of rotavirus-infected patients showed signs of fever, which on average lasted for 2.2 days. Interestingly, fever was found to be much more frequent in rotavirus-infected patients compared to patients with enteric adenovirus (44%) or bacterial gastroenteritis (69%). In another Scandinavian study comprising 118 rotavirus-infected children [75], elevated body temperature was measured in 86–90% of the patients (depending on subtype), with a mean duration of 2.1–2.3 days. Of these children, 47% were in the range of 37.7–38.9 °C and 43% had temperatures above 39 °C. 

An American study showed that children suffering from acute rotavirus gastroenteritis exhibit higher levels of IL-6 in serum when also experiencing fever, and children with both fever and more episodes of diarrhea also exhibit higher levels of TNF-α [81]. The study also found that the levels of IL-6, IL-10, and IFN-γ were significantly higher in children with acute rotavirus infection compared to healthy children without diarrhea [81]. 

Another study, from Turkey, comparing bacterial and rotavirus gastroenteritis found that elevated IL-6 and TNF-α was lower in rotavirus-infected children compared to children with bacterial gastroenteritis [80]. Rotavirus-infected children also had slightly elevated fever compared to bacterially infected children and did not have any C-reactive protein (CRP), leucocytes, or blood in their feces. It should be noted that these differences in cytokine responses during rotavirus infection are detectable when compared to bacterial infections. Standalone, cytokine and inflammatory elevations during rotavirus infection is mostly modest [10,15,16,17,18]. Little is known about the role of cytokines in the pathogenesis of rotavirus disease and needs to be further investigated.

The temperature homeostasis in mammals is regulated by neurons located in the preoptic area of the hypothalamus, which function as a thermostat that keeps the body temperature within an optimal preset range, despite varying ambient temperature [82,83]. These neurons can respond to nervous input, cytokines, or circulating hormones like prostaglandin, and slightly raise this setpoint to increase the efficacy of the immune response during a pathogenic invasion [84,85]. Although acutely effective, sustained high temperature can rapidly become harmful and even fatal [86]. 

Depression, loss of appetite, and anorexia, which reduce the overall metabolic rate, are also common during fever [58,87]. This is rather counter-intuitive, as a systemic increase of body temperature is extremely costly and requires a 10–12.5% increase of metabolic rate for a single degree Celsius increase of body temperature [79]. However, the fact that fever has been retained throughout vertebrate evolution strongly suggests that febrile temperatures confer a survival advantage. 

The induction and maintenance of fever during infection involves a tightly coordinated interplay between the innate immune system and neuronal circuitries within the central and peripheral nervous systems. Acetylcholine contributes to fever by stimulating muscle myocytes to induce shivering, and norepinephrine elevates body temperature by increasing thermogenesis in brown adipose tissue as well as inducing vasoconstriction to prevent passive heat loss [79]. Release of prostaglandin E2 (PGE2) as well as pyrogenic cytokines like IL-1, IL-6, and TNF-α, from innate immune cells, can also induce fever [88]. Where and how these systems are perturbated during rotavirus infection remain elusive. 

### 4.4. Fatigue and Sleepiness

Fatigue appears central in sickness behavior because it is a strong signal to rest and focus the energy to combat infection or inflammation and regain health. Fatigue is a result of the modification of metabolic pathways through an endocrine loop, initiated in the hypothalamus and amplified by the pituitary gland. The release of corticotrophin-releasing hormones, cortisol, and adreno-corticotropic hormones affect nutrient homeostasis in tissues such as the liver, muscles, and adipose tissue. A meta-analysis of human sickness behavior showed that fatigue was one of the most commonly reported symptoms and is associated with IL-6 and IL-1 [89].

Fatigue should not be confused with sleepiness, i.e., sleep propensity, which is the normal signal for sleep, and although sleepiness and fatigue are often used interchangeably, they are two distinct responses [90]. Sleepiness is a driving force related to the need for sleep, while fatigue is a more general signal to avoid activity [90]. 

Sleep is altered during infection, but it is unknown why [91]. However, it is known that infection-induced alterations in sleep is a CNS response to IL-1 and TNF-α. Neurons immunoreactive for IL-1 and TNF and involved in the regulation of sleep and wake behavior are notably located in the hypothalamus, but also in the hippocampus and the brainstem [91]. It is known that the regulation of body temperature is coupled to sleep and the changes in sleep architecture that occur during infection, suppressed REM (rapid eye moment) sleep and increased but more fragmented non-REM sleep, might have fever-promoting benefits [91]. Sleep is proposed to be an acute-phase response and a sickness behavior that promotes recovery and increases survival. Information about fatigue is not easily accessed in children but in a rotavirus outbreak study among college students, 50% of the 83 students with rotavirus infection declare that they experienced fatigue [2]. 

### 4.5. Stress

The sympathetic system works as fight or flight response and parasympathetic activation induces the rest response. Rotavirus infection, like other viral pathogens, cause infection-induced stress response, likely by activation of the sympathetic pathway to the hypothalamus and the release of cortisol. Activation of the hypothalamic–pituitary–adrenal (HPA) axis is well known to subserve the body’s response to a stressor, and viral infections, in general, are physiologically stressful [88]. 

Catecholamines, the end-product of sympathetic nervous system activation, and glucocorticoids orchestrate the “fight or flight” response, with rapid mobilization of energy to critical muscles and the brain, concomitant with increased heart rate, blood pressure, and breathing rate to facilitate rapid transport of nutrients and oxygen to relevant tissues. At the same time the HPA axis also assists in shunting metabolic resources from growth, digestion, reproduction, and certain aspects of immunity to the more immediate and acute functions.

The physiological function of stress–induced increases in glucocorticoid levels is to protect against the normal defense reactions, e.g., immune response/inflammation that are activated by stress; glucocorticoids accomplish this function by turning off those defense reactions, thus preventing them from overshooting. This can be seen as the glucocorticoid actions may help mediate the recovery from the stress response, rather than mediate the stress response itself. In a systematic review of 15 studies of acute illness [92], it was found that cortisol levels were more than 3-fold higher in the group with severe gastroenteritis than in the control group [92]. Activation of the HPA axis is well known to subserve the body’s response to a stressor, and viral infections, in general, are physiologically stressful, as indicated by the concomitant activation of the HPA axis [88,93].

### 4.6. Loss of Appetite

Loss or lack of appetite, also referred to as anorexia, are common during infection conditions, and have been reported as high as 83% in an rotavirus outbreak among college students [2]. The brain continuously receives information from the periphery regarding energy stores and energy needs and processes this information in order to regulate feeding behavior [94]. The brain also senses and responds to peripheral infection and inflammatory processes, which in turn, affect feeding and metabolism, indicating that during inflammation and or infection the immune response largely depends on the energy status of the host [94]. Understanding the mechanisms through which the brain regulates appetite and feeding behavior will provide insights into the metabolic adaptation for therapeutic intervention [94].

Reducing the level of food intake is a good strategy to rest and enable evacuation of existing pathogens from the gastric system. It also reduces the risk of introducing new pathogens. This might be perceived as counter-intuitive, as access to nutrients is crucial for successfully combating an infectious pathogen. 

The hypothalamic melanocortin system is heavily involved in the regulation of appetite, and has been considered as a promising target to control appetite during disease conditions [94]. Another key region controlling appetite is the dorsal vagal complex (DVC), a brain structure that comprises the area postrema (AP), the nucleus of the solitary tract (NTS) and the dorsal motor nucleus of the vagus (DMV) [94]. Several peripheral mechanisms, including motor functions of the stomach as well as released peptides and hormones, provide feedback to the hypothalamic circuitry and vagal complexes of the brain to regulate appetite and thereby balance consumed and expended body energy [95]. There are cytokines, and prostaglandin E, involved in appetite regulation and these signals can reach the brain through vagal sensory nerve receptor or by the circulation [94].

Although several microorganisms are known for their ability to manipulate host defenses to their own advantage, the role of anorexia induced by pathogens is not clear yet. It may also be that access to nutrition can have different effects depending on the pathogen itself. While fasting appear protective during bacterial infection, it seems to have detrimental effects during viral infection, highlighting divergent metabolic requirements [96]. Low nutrition status has been associated with risk for more severe rotavirus infection, however not yet proved and studies have shown opposing results [97,98]. However, in children with metabolic disorders, rotavirus infection was associated with increased morbidity and mortality [99]. Weather if the host response of reduced appetite with a fasting state promotes rotavirus recovery, or if nutrition supplement is beneficial, needs to be further investigated.

## 5. Evolution of Defense Strategies

Characteristics that benefit our survival have been preserved during evolution. Humans and animals evolved with viruses, and up to 2/3 of the human genome is derived from viruses and transposable elements [100]. Early in human evolution, the means to properly store food was lacking, and food quality was uncontrolled. The ability to handle ingestion of toxic and pathogenic agents was literally a matter of life and death. 

Successful host defense strategies against viral pathogens were, and to large extent still are, strongly favorable for the survival of the species and thus evolutionary conserved [58]. These facts have led to the evolution of complex physiological and behavioral defense strategies against ingulfed harmful substances, like rotavirus.

Defensive responses like fever, sleepiness and loss of appetite are not weaknesses due to the infection per se, but rather an acquired adaptive strategy for survival [58]. Counter-intuitively, viral infections, which normally enter cells, are considered more likely to induce vomiting, than bacterial infections, which normally do not enter cells [64]. It should, however, be noted that the mechanism underlying rotavirus vomiting is at least partly driven by the NSP4 enterotoxin, and in an essence similar to the mechanism of toxin producing bacteria that also induce vomiting [101], indicating that the neuronal pathways have evolved against toxins rather than specific pathogenic agents. Interestingly, virus is the Latin word for poison. 

Gastrointestinal bacterial infections usually cause prolonged bloody diarrhea [64]. Acute viral gastroenteritis, on the other hand, is characterized by watery diarrhea and a low inflammatory response with mild elevation of serum inflammatory markers [102]. These characteristics are also recapitulated in the symptomology as viral gastroenteritis are more acute, but resolve faster than bacterial infections [102]. 

## 6. Conclusions

The question of whether sickness symptoms are harmful or protective during rotavirus gastroenteritis has no straightforward answer. It is case-specific and depends on many variables. From an evolutionary point of view, sickness symptoms are indeed manifestations of survival strategies and adaptive host defensive responses against different kinds of danger. This should be considered, particularly when prescribing therapies and treatments against sickness symptoms, rather than the pathogen and its toxic components. While it is possible in modern medicine to identify specific pathogens like rotavirus and provide targeted therapies, adequate knowledge about the body defense mechanisms is in many aspects largely lacking.

Blocking sickness symptoms, without adequate knowledge about the underlying mechanisms, could have negative impact, introduce adverse effects, and lead to prolonged recovery. In cases when a therapy does not have an obvious and direct benefit, it might thus be better to let the host carry on its evolved adaptive responses, despite any temporary discomfort that might arise. At the same time, excessive defensive host measures could also become harmful. Such is the case with prolonged rotavirus diarrhea and vomiting, which arguably provide little to no benefit to the host during the later stages of the disease but pose an imminent risk of fatal outcome. In such a case, blocking the sickness symptom is directly beneficial and should be considered.

## Figures and Tables

**Figure 1 viruses-16-01086-f001:**
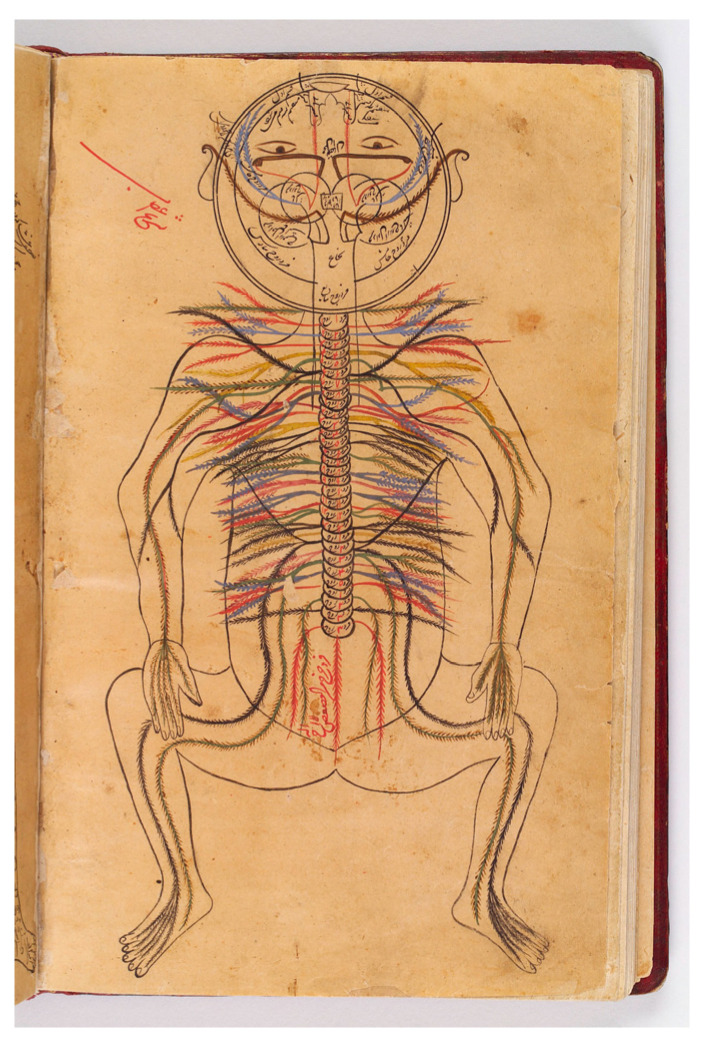
The nervous system innervates the peritoneum. An illustration of the nervous system of the human body by Ibn Sina (Avicenna) in Qānūn fī al-Tibb (the Canon of Medicine). Image available on the internet, open source, courtesy of the Wellcome Collection (Link: https://wellcomecollection.org/works/mx97zpqj [accessed on 4 June 2024]).

**Figure 2 viruses-16-01086-f002:**
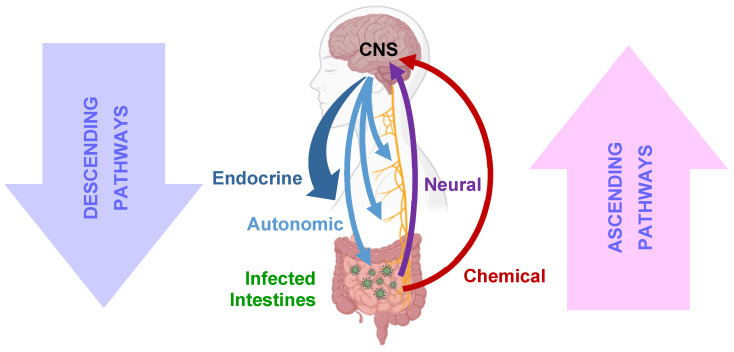
Bidirectional gut–brain crosstalk can occur via multiple independent chemical and electrical pathways. During rotavirus infection, the information of the pathogenic presence can reach the central nervous system (CNS) through ascending neural pathways, which include the vagus nerve and the spinal pathways. Released chemicals, including peptides, hormones and toxins, can also carry information and reach the brain through vascular or lymphatic systems. The brain processes these signals and coordinates the defense and output back to the periphery through descending pathways. While the autonomic nervous system, consisting of the sympathetic and the parasympathetic systems, provides direct, specific, and rapid access to multiple organs, the endocrine system, by releasing circulating hormones, performs the same role for broader systemic regulation at slower speed.

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
