# Peer review of "Rotavirus Sickness Symptoms: Manifestations of Defensive Responses from the Brain"

_viruses, 2024, doi:10.3390/v16071086_

Round 1

Reviewer 1 Report

Comments and Suggestions for Authors

In this review, the authors explore the mechanisms responsible for the symptoms and behavioural modifications during rotavirus infections. Additionally, they examine both experimental and clinical data to elucidate the evolutionary impact of these responses to these diseases. This discussion is vital in the field and this review will provide a deep understanding of how rotavirus interacts with its hosts and the potential evolutionary advantages these effects might lead to. The review is well written and the figures are well structured. The writing is clear and it can serve as stepping stones in the field. I strongly suggest the publication in Viruses.

Author Response

Dear reviewers,

We appreciate for the comments particularly with reviewer one, who emphasize that we have raised aspects that has not been discussed previously regarding rotavirus symptomology.

Sincerely,

Marie Hagbom, PhD, associate professor

Reviewer 2 Report

Comments and Suggestions for Authors

In this manuscript, the authors discuss clinical symptoms and behaviors after rotavirus infection.  They conclude that some of the symptoms and behaviors are protective against rotavirus infection.  The manuscript is written in good English.  However, the manuscript is quite long and most of the parts are unrelated to rotavirus infection.  In addition, it is well known that clinical symptoms after viral infection are generally protective against viruses.  So, this manuscript is not informative in the current form.  The authors should focus on the parameters that should be optimized during medical treatment of rotavirus infection and formulate a guideline for treatment of rotavirus infection considering protective responses of patients’ bodies and currently available medical strategies.  

Author Response

(The authors gave the same response as above.)
